# Immune Checkpoint Inhibitors in People Living with HIV/AIDS: Facts and Controversies

**DOI:** 10.3390/cells10092227

**Published:** 2021-08-27

**Authors:** Valeria Castelli, Andrea Lombardi, Emanuele Palomba, Giorgio Bozzi, Riccardo Ungaro, Laura Alagna, Davide Mangioni, Antonio Muscatello, Alessandra Bandera, Andrea Gori

**Affiliations:** 1Infectious Diseases Unit, Foundation IRCCS Ca’ Granda Ospedale Maggiore Policlinico, 20122 Milano, Italy; valeria.castelli@unimi.it (V.C.); emanuele.palomba@unimi.it (E.P.); giorgio.bozzi@policlinico.mi.it (G.B.); riccardo.ungaro@policlinico.mi.it (R.U.); laura.alagna@policlinico.mi.it (L.A.); davide.mangioni@policlinico.mi.it (D.M.); antonio.muscatello@policlinico.mi.it (A.M.); alessandra.bandera@unimi.it (A.B.); andrea.gori@unimi.it (A.G.); 2Department of Pathophysiology and Transplantation, University of Milano, 20122 Milano, Italy; 3Centre for Multidisciplinary Research in Health Science (MACH), University of Milano, 20122 Milano, Italy

**Keywords:** immune checkpoint inhibitors, HIV, immune exhaustion

## Abstract

Immune checkpoint inhibitors (ICIs) are reshaping the landscape of cancer treatment, redefining the prognosis of several tumors. They act by restoring the cytotoxic activity of tumor-specific T lymphocytes that are in a condition of immune exhaustion. The same condition has been widely described in chronic HIV infection. In this review, we dissect the role of ICIs in people living with HIV/AIDS (PLWHIV). First, we provide an overview of the immunologic scenario. Second, we discuss the possible use of ICIs as adjuvant treatment of HIV to achieve elimination of the viral reservoir. Third, we examine the influence of HIV infection on ICI safety and effectiveness. Finally, we describe how the administration of ICIs impacts opportunistic infections.

## 1. Introduction

The landscape of oncologic treatments is undergoing an epochal change of perspective since the advent of therapeutic regimens based on immune checkpoint inhibitors (ICIs). These molecules act on the immune system, re-establishing the activity of cytotoxic T lymphocytes (CTLs) against neoplastic cells. Therefore, they are defined as cancer immunotherapies, and the daily-growing field where they are employed has been called immuno-oncology.

The first approved ICI was ipilimumab, an anti-cytotoxic T-lymphocyte-associated protein 4 (CTLA-4), for the treatment of unresectable or metastatic melanoma in 2011 [1]. Since then several other drugs of this class have been approved by regulatory agencies, all targeting the programmed cell death protein 1 (PD-1)/programmed death-ligand 1 (PD-L1) pathway: atezolizumab [2], avelumab [3], cemiplimab [4], dostarlimab [5], durvalumab [6], nivolumab [7], and pembrolizumab [8]. ICIs currently approved by the Food and Drug Administration (FDA) with their indications are shown in Table 1.

The reason for their widespread application resides in their innovative therapeutic approach, which has guaranteed excellent results against cancers once considered less susceptible to treatment. For example, 20 years ago the average life expectancy for patients with advanced melanoma was 6/7 months, whereas recent studies have shown an overall survival at 5 years in 52% of patients treated with nivolumab-plus-ipilimumab and 44% of patients treated with nivolumab alone [9]. Similar excellent results have been described for non-small-cell lung cancer and advanced urothelial carcinoma [10,11,12].

Their own mechanism of action, which has led to their success in certain malignancies, has hampered their use for cancer treatment among people living with HIV/AIDS (PLWHIV). Indeed, the alteration of T cell immune response described during chronic HIV infection resembles the immune exhaustion observed among oncologic patients. Several doubts arose about the risk of eliciting an immune response against host cells infected by HIV. Consequently, PLWHIV were and still are excluded from many randomized controlled trials (RCT) investigating ICIs for neoplastic conditions [13]. Table 2 reports the studies registered on Clinicaltrial.gov assessing the efficacy of ICIs for treatment of cancers specifically in PLWHIV.

The aim of this review is to dissect the role of ICIs in PLWHIV. First, we provide an overview of the immunologic background. Second, we discuss the possible use of ICIs as adjuvant treatment of HIV to achieve elimination of the viral reservoir. Third, we examine the influence of HIV infection on ICIs safety and effectiveness. Finally, we describe how the administration of ICIs impacts opportunistic infections.

## 2. The Immunologic Background

T cells exhaustion is a condition that occurs in chronic infections and in several cancers. Initially it was identified and described in a murine model of infection by lymphocytic choriomeningitis virus [14,15,16], but subsequently was depicted also in other infections [17] and many tumors [18]. Several mechanisms lead to T cell exhaustion: cell-to-cell signals including prolonged T cell receptor (TCR) engagement and co-stimulatory and/or co-inhibitory signals, soluble factors such as excessive levels of inflammatory cytokines, and tissue and microenvironmental influences [19]. As a consequence of these stimuli, T cells progressively develop a state of exhaustion characterized by the inability to elaborate the arrays of effector functions associated with typical effector and memory T cells. This dysfunction can lead to clonal deletion of antigen-specific T cells [20]. The result is the inability of specific T cells to control and eliminate infected or neoplastic cells. The prolonged and/or high expression of multiple inhibitory receptors is a key feature of T cell exhaustion. Among these, CTLA-4 and PD-1 are the inhibitory receptors more involved in T cell exhaustion. CTLA-4 and PD-1 propagate inhibitory signals that converge on Akt to limit cellular metabolism. Whereas PD-1 disrupts the intracellular accumulation of 3-phosphorylated phosphatidylinositol lipids, CTLA-4 targets downstream effectors of PI3K through activation of the serine/threonine phosphatase PP2A [21]. The administration of molecules able to block these receptors or their ligands has been shown to be able to restore antiviral or antitumoral activities in several experimental models [22].

Progression of HIV infection occurs in most patients in the presence of persistently high viremia and is associated with a loss of immune control of viral replication. Virus-specific CD8 T cells partially suppress HIV viral replication in the initial stages of infection [23], but, with persistently high levels of viral antigens, HIV-specific T cells become exhausted and lose their capacity to efficiently kill the infected cells. In addition to high levels of circulating viral antigens, the strong pro-inflammatory immune activation during HIV infection and the T-cell subpopulation imbalance during HIV infection contribute to the development of T-cell exhaustion [24,25].

Combination antiretroviral therapy (cART) has a dramatic effect on HIV infection. After treatment start, the majority of patients experience a significant decline in viral load; the aim is reaching undetectability, and, with time, the count of CD4+ T cells can increase. Unfortunately the drugs currently available cannot eradicate the infection, but only control it, which is associated with significant clinical benefit [26]. Patients receiving efficacious cART display a progressive reduction of ICI expression on T cells, while their values remain higher than those observed in HIV-uninfected individuals [23]. The best current strategy for fully successful immune restoration is early cART initiation, which can prevent acquired immunodeficiency syndrome (AIDS)-associated events and restrict cell subset imbalances and dysfunction, while preserving the structural integrity of lymphoid tissues [27].

PD-1 is a crucial immune checkpoint in chronic HIV infection and is significantly upregulated on HIV-specific CD8 T-cells in patients naïve to antiretroviral therapy. Its expression correlates with impaired HIV-specific CD8 T-cell function as well as predictors of disease progression positively with plasma viral load and inversely with CD4 T-cell count [28]. Importantly, the level of PD-1 surface expression is the primary determinant of apoptosis sensitivity of HIV-specific CD8+ T-cells. Interestingly, cytomegalovirus (CMV)-specific CD8+ T cells from the same individuals whose HIV-specific CD8+ T-cells upregulate PD-1, do not upregulate PD-1 and maintain the production of high levels of cytokines, suggesting the specific impact of HIV [29]. Likewise, PD-1 expression on CD4+ T cells is directly correlated with viral load and inversely correlated with CD4+ T-cell count. The blockade of the PD-1/PD-L1 pathway can restore both HIV-specific CD4+ and CD8+ T-cell function, suggesting that this pathway is operative during persistent viral infection and is a reversible defect in HIV-specific T-cell function [30].

Similarly to PD-1, several pieces of evidence highlight an essential role also for CTLA-4 in driving HIV-specific T cells toward exhaustion. Initial works identified how CTLA-4 is more highly expressed by unstimulated blood CD4+ T cells from HIV patients than by control T cells [31]. In their pivotal work, Kaufmann and colleagues confirmed how CTLA-4 is selectively upregulated in HIV–specific CD4+ T cells but not in CD8+ T cells in various categories of HIV-infected subjects (untreated HIV-infected subjects, viremic controllers, untreated chronically infected subjects, subjects with acute untreated HIV infection), excluding elite controllers. CTLA-4 expression correlates positively with disease progression and negatively with the capacity of CD4+ T cells to produce interleukin 2 in response to viral antigen, and consequently to clear the infection. Most HIV-specific CD4+ T cells co-express CTLA-4 alongside PD-1. Interestingly, in vitro blockade of CTLA-4 augmented HIV-specific CD4+ T cell function [32].

Alongside PD-1 and CTLA-4, other immune checkpoints associated with T cell immune exhaustion in PLWHIV are T cell immunoglobulin and mucin domain-containing protein 3 (TIM-3), lymphocyte-activation gene 3 (LAG-3), and T cell immunoreceptor with Ig and ITIM domains (TIGIT). Levels of TIM-3 expression on T cells from HIV-infected individuals correlate positively with HIV-1 viral load and inversely with CD4+ T cell count. In progressive HIV infection, TIM-3 expression is upregulated on HIV-specific CD8+ T cells. TIM-3-expressing T cells do not produce cytokine or proliferate in response to antigen and exhibit impaired Stat5, Erk1/2, and p38 signaling. The blockade of the TIM-3 signaling pathway can restore proliferation and enhance cytokine production in HIV-specific T cells, suggesting how this inhibitory receptor is central in the immune response against HIV [33]. Moreover, the downregulation of TIM-3 is associated with restoration of CD4+ T cell counts in PLWHIV receiving cART with suppressed viremia [34]. Similarly, HIV-1 infection results in a significant increase in LAG-3 expression in both the peripheral blood and the lymph nodes, and this upregulation is dramatically manifested on both CD4+ and CD8+ T cells and correlates with disease progression. Prolonged cART reduces the expression of LAG-3 on both CD4+ and CD8+ T cells, and the ex vivo blockade of LAG-3 significantly augments HIV-specific CD4+ and CD8+ T cell responses, whereas the overexpression of LAG-3 in T cells or the stimulation of LAG-3 on T cells leads to the reduction of T cell responses. Overall, these data show how the LAG-3/MHC class II pathway plays an immunoregulatory role, thereby providing an important target for enhancing HIV-specific responses in infected patients [35]. During HIV infection, CD8+ T cells exhibit higher levels also of TIGIT. Increased frequencies of TIGIT+ and TIGIT+/PD-1+/CD8+ T cells correlate with parameters of HIV disease progression. TIGIT remain elevated despite viral suppression both in individuals with pharmacological antiretroviral control and in elite controllers. Ex-vivo single or combinational antibody blockade of TIGIT is able to restore viral-specific CD8+ T cell effector responses. The frequency of TIGIT+/CD4+ T cells correlates with CD4+ T cell total HIV DNA. Overall, these findings identify TIGIT as another marker of dysfunctional HIV-specific T cells and suggest how TIGIT along with other checkpoint receptors may be a new possible target in HIV treatment to reverse T cell exhaustion [36].

Other than immune checkpoints, inflammatory cytokines also participate in the process of immune exhaustion. Among them, IL-10 has a crucial role. IL-10 mRNA expression is increased in the setting of chronic uncontrolled HIV infection and correlates with plasma viremia in infected persons. IL-10Rα blockade restores not only HIV-specific CD4 cell proliferation but also antigen-specific CD8 T-cell proliferation. IL-10 mRNA expression and IL-10 plasma levels are reduced through successful antiretroviral treatment, indicating a direct effect of viral antigen load on IL-10 production. Overall, IL-10 contributes to a reversible T-cell dysfunction in HIV infected persons, and its levels are directly correlated to viral antigen levels [37]. Figure 1 provides a summary of the immune exhaustion process occurring in PLWHIV.

## 3. The Impact of ICIs on HIV Replication

PLWHIV and patients with other chronic infections were excluded from the first trials that validated the use of ICIs in clinical practice, based on the concern that immune checkpoint inhibition could lead to a surge of immune response against the virus. In the following years, several studies investigated this hypothesis, as viral reactivation could both represent a severe adverse reaction and, on the other hand, prove pivotal for future eradication strategies.

ICIs, namely ipilimumab for the treatment of metastatic melanoma, were first prescribed to a patient living with HIV in 2011, as described by Burke et al. [38]. Since then, several case series, retrospective cohort studies, and systematic reviews have accumulated. Six years later, PLWHIV were first included into a prospective trial with an ICI-based intervention: two HIV patients received nivolumab for squamous cell carcinoma of the anal canal [39]. Neither of the participants reported serious immunologic or virologic events after exposure. Since 2017, other trials—some of them still ongoing—have been opened to HIV-positive participants. In 2019, in a systematic review of PLWHIV with cancer undergoing treatment with ICIs, Kim and Cook concluded that PLWHIV treated with ICIs achieve therapeutic outcomes similar to HIV-negative patients and did not report significant virologic or immunologic events [40]. A French multicenter retrospective observational study of PLWHIV with various oncologic conditions (2014–2019) collected data on 23 participants with undetectable HIV RNA treated with pembrolizumab and nivolumab (mostly for NSCLC): all patients who continued receiving cART during treatment did not experience loss of viremia control [41]. In conclusion, treatment with ICIs does not seem to have deleterious effects on HIV infection. Randomized controlled trials are needed to further validate such observations and optimize cART during therapy with ICIs [42].

Another line of research involving PLWHIV and ICIs explores the use of the latter for treatment strategies aimed at viral eradication. As of today, strategies allowing viral control in the absence of cART are not available in clinical practice. Upon diagnosis, PLWHIV therefore become candidates for life-long chronic treatment, posing adherence, toxicity, availability, and cost issues. On this basis, vast effort has been put into investigating functional cure strategies. The main obstacle to permanent viral suppression is posed by the HIV reservoir, constituted by cells latently infected with integrated proviral genomes [43]. Various pre-clinical studies have shown that inducible, replication competent HIV genomes are preferably harbored in CD4+ cells over-expressing immune checkpoints, and that immune checkpoint expression correlates with total HIV DNA [44,45]. In vitro, ICIs have been shown to restore HIV-specific CD8+ cells and to increase HIV production from reservoir cells [29].

Due to these findings, ICIs have been recognized as candidates for the shock and kill strategy, that is, using certain drugs to activate the latent reservoir (“shock”) to enable HIV-specific immunity to recognize and eliminate (“kill”) the cells that harbored latent HIV, effectively purging the reservoir [46]. Despite promising results in preclinical studies and animal models, the impact of this strategy on the human host is yet to be fully defined. In 2015, a case report on ipilimumab used against melanoma in an HIV-positive patient was published, and for the first time in the literature, the effect of ICI treatment on the HIV reservoir was described [47]. A second case reported a decrease in HIV DNA levels and a simultaneous plasmatic HIV RNA peak after the use of nivolumab (used for NSCLC) in an HIV-positive patient [48]. Abbar and collaborators performed a systematic review of the available literature in 2020, rounding up 176 PLWHIV subjected to ICI treatment, some of whom had HIV DNA and/or HIV-specific CD8+ responses measured [49]. Among individuals with both molecular and immunological variables available, no effect was observed in more than a half, and only one patient displayed both an HIV-specific CD8+ cell increase and HIV DNA decrease. Despite the underlying safety viro-immunologic profile of ICIs, the authors concluded that such drugs have a limited impact on the HIV reservoir. However, according to various ongoing trials, ICIs remain suitable candidates as combination drugs in synergistic strategies towards HIV cure. Probably these strategies will employ latency reversing agents, and preliminary results suggest the ability of anti-PD1 molecules to potentiate HIV latency reversal in cART suppressed PLWHIV [50].

## 4. Safety and Efficacy of ICIs in PLWHIV

Even though PLWHIV are at increased risk of developing several cancers, this population has been historically excluded from clinical trials assessing the efficacy of ICIs, thus inhibiting broad implementation of these therapies among PLWHIV with cancer [51]. As stated above, this exclusion was due to the fear of side effects related to reestablishing the immunity against HIV and to the concern that this population lacks a sufficient underlying T-cell immunity to benefit from therapy. In the last few years, encouraging trends in the inclusion of PLWHIV in ICI trials have been identified [51,52]. These data have already been collected in a small number of reviews [40,41,42] assessing the safety and efficacy of ICIs in this subgroup of patients with advanced cancer. In PLWHIV, ICIs have been employed for the treatment of both AIDS-defining malignancies [52] and non-AIDS defining malignancies.

In their systematic review, Kim and Cook [40] described 73 PLWHIV treated with ICIs. Most patients were reported to have non-AIDS defining malignancies: 25 patients had non-small cell lung cancer (NSCLC) and 16 patients had melanoma, whereas nine patients had Kaposi sarcoma. Pre-treatment and post-treatment HIV loads were available in 34 of 73 patients. Twenty-eight individuals had undetectable viral load. HIV remained suppressed in 26 of the 28 (93%) with undetectable HIV load, and their CD4 cell counts increased (mean [SD] change, 12.3 [28.5]/μL). This data could suggest that ICI treatment does not impact HIV control, maintaining undetectable HIV viral load and not causing CD4 T cell counts to decrease. Unfortunately, data on HIV DNA were not available to assess the impact of ICIs on the viral reservoir. Immune checkpoint inhibitor therapy was generally well tolerated, with grade 3 or higher immune-related adverse events noted in six of 70 patients (8.6%), a value similar to that observed in patients without HIV infection [40].

Uldrick et al. [53] conducted an open-label study to assess the safety of pembrolizumab in PLWHIV with advanced cancer. Eligibility criteria were HIV infection, a concomitant advanced cancer, a CD4+ T-cell count greater than or equal to 100 cells/μL, cART for 4 or more weeks, and an HIV viral load of less than 200 copies/mL. Overall, 30 PLWHIV were treated with at least one cycle of pembrolizumab, monitoring viro-immunological status, adverse events, and disease progression. The majority of treatment-emergent adverse events were grade 1 or 2 and included anemia, fatigue, nausea, and hypothyroidism. Based on this study, the authors concluded that pembrolizumab has a similar safety profile in PLWHIV with suppressed HIV or low-level viremia on cART and advanced cancer to that observed in the general population. Interestingly, one participant with detectable Kaposi sarcoma (KS) herpesvirus (KSHV) viremia before receiving pembrolizumab developed a polyclonal KSHV-associated B-cell lymphoproliferative disorder and died, suggesting the need for additional monitoring in patients with this active infection while receiving ICIs. Regarding viro-immunological status, pembrolizumab does not appear to influence CD4+ T-cell counts or HIV viral load. The impact on viro-immunological status reported by Uldrik et al. is in accordance with data previously reported in the literature. In a case series [54] of three patients with HIV infection affected by Merkel-cell carcinoma, pembrolizumab administration seemed not to modify viro-immunological status, with all patients having an HIV viral load consistently undetectable at baseline and after treatment, maintaining stable CD4+ T cell counts during treatment with ICI. Similarly, in a retrospective study [55] that collected data from nine patients with HIV infection and Kaposi sarcoma treated with ICIs, eight patients received nivolumab and one pembrolizumab. At baseline, all patients were receiving antiretroviral therapy, with well-controlled HIV viral load in seven of the nine patients. Seven patients (78%) during ICI treatment experienced an improvement in CD4+ T cell counts.

Regarding efficacy of ICIs in PLWHIV, Uldrik et al. [53] documented a protocol-defined clinical benefit in 17% of participants affected by different type of cancers, suggesting a comparable efficacy to the HIV-negative population with advanced cancer. In agreement with this preliminary data, Bari et al. [56] performed a retrospective analysis of 17 HIV patients treated with one of the PD-1/PD-L1 inhibitors for advanced cancer: 10 patients had lung cancer, two hepatocellular cancer, two anal cancers, one kidney cancer, one non-Hodgkin’s lymphoma, and one advanced basal cell carcinoma. Ten patients responded to treatment, of whom five had partial response and five had stable disease. Matching retrospectively reported evidence and Uldrik’s prospective study, anti-PD1/PD-L1 therapy appears to be safe and effective in HIV patients with cancer. However, these data are based on small samples. Further prospective studies on the use of ICIs in PLWHIV with advanced cancer are currently ongoing [57,58,59] to better assess safety and efficacy and to promote broader use of this immunotherapy in PLWHIV with stable immunovirological status.

## 5. ICIs and Opportunistic Infections

The use of ICIs is associated with immune-related adverse effects (irAEs) related to the upregulated immune system. These toxicities can affect a variety of organs including lung (pneumonitis), gastrointestinal tract (colitis), liver (hepatitis), skin (rash), pancreas (pancreatitis), endocrine (hypophysitis, thyroiditis), and kidneys (nephritis). Standard management algorithms from scientific societies [60] recommend the use of immunosuppressive medications such as steroids or, in case of refractory disease, tumor necrosis factor alpha (TNF-α) inhibitors to manage these immune-related adverse events. The immunosuppression deriving from the use of these drugs has been reported to be a risk factor for opportunistic infections [61,62].

Several cases of opportunistic infections among patients with melanoma receiving the CTLA-4 inhibitor ipilimumab have been published, including invasive aspergillosis [63], cytomegalovirus-induced hepatitis [64], and pneumocystis pneumonia (PJP) [65].

One retrospective study has systemically evaluated the risk of infection in 748 melanoma patients receiving ICIs (CTLA-4, PD-1, and/or PD-L1) [66]: 658 (73.2%) received ipilimumab, 52 (5.7%) received nivolumab, 83 (9.2%) received pembrolizumab, and 105 (11.7%) received a combination therapy. Serious infections developed in 54 patients (7.3%), including bacterial pneumonia, intra-abdominal infection, invasive pulmonary aspergillosis, pneumocystis pneumonia, disseminated herpes zoster (HZ), cytomegalovirus colitis, and *Strongyloides stercoralis* hyperinfestation syndrome. The major risk factors for infection were use of corticosteroids and/or infliximab and a combination of ipilimumab with nivolumab.

Another retrospective study [67] analyzed 167 patients affected by non-small cell lung carcinoma (NSCLC) treated with nivolumab. Thirty-two (19.2%) out of 167 patients developed serious infections, most of them bacterial (78.1%), while 18.8% were viral and 6.3% fungal; a case of pulmonary invasive aspergillosis and two cases of herpes zoster reactivation have been reported.

Various cases of reactivation of latent tuberculosis (LTBI) among patients treated with nivolumab or pembrolizumab have been described in literature in patients not receiving further immunosuppressive therapies for irAEs. In 2016 Lee et al. [68] described the first case of LTBI reactivation in a patient undergoing immunotherapy with pembrolizumab in Hodgkin’s lymphoma. In some recent reviews [69,70], reactivations or primary infections by M. tuberculosis have been described in patients treated with ICIs for various neoplastic diseases (non-small cell lung cancer, melanoma, Hodgkin’s lymphoma, nasopharyngeal cancer, squamous cell carcinoma of the oral cavity, Merkel cell carcinoma). The risk of primary infection or reactivation during ICI treatments could be influenced by the underlying malignant disease causing local and systemic immune dysregulation. Particularly, immune checkpoint inhibitors’ immunomodulatory effect may enhance the immune reaction to infectious disease, resulting in reactivation and unmasking of latent infections. An immune reconstitution inflammatory syndrome may explain previously reported cases, such as the progression of chronic progressive pulmonary aspergillosis in a patient affected by lung adenocarcinoma treated with nivolumab in Japan [71].

As for previously reported infections, ICIs are not directly associated with an increased risk of PJP. A recent meta-analysis [72] investigating the incidence and relative risk (RR) of lung infections observed in phase II/III studies including patients with solid tumors undergoing treatment with ICIs did not show an increased RR compared to the standard chemotherapy. However, as previously described, treatment of irAEs may lead to an increased risk of PJP. Del Castillo et al. [66] described 3 cases of PJP in patients treated with corticosteroids and infliximab. Seven additional clinical cases described the development of PJP as a consequence of the use of immunosuppressants for the management of irAEs. Four patients were affected by melanoma [65,73,74], four by lung cancer [75], and one by Hodgkin’s lymphoma [73]. In a retrospective study of 515 patients treated with ICIs for malignancy, 23 required treatment with high doses of corticosteroids secondary to the development of immune-mediated pneumonia, six (26%) of whom developed PJP (all in the absence of specific prophylaxis for PJP); in this study the presence of lymphopenia (lymphocytes < 755 cells/µL) was a predisposing factor for the development of PJP during steroid treatment [76]. Few cases of PJP associated with treatment with ICIs in the absence of previous or concomitant immunosuppressive treatment for irAEs have been described in the literature; the majority of patients presented significant comorbidities or anamnestic history of chemotherapy or radiotherapy treatments with a possible etiological role in the development of opportunistic infections [77].

The burden of opportunistic infections in PLWHIV depends on their viro-immunological status. Recovery of immunological competence has been made possible with the advent of cART. As is widely known, today the commonest opportunistic infections of the pre-cART era can be avoided in PLWHIV with suppressed viral load and CD4 cell counts > 200/mmc.

Currently, no cases of opportunistic infections have been reported in patients with HIV and cancer treated with ICIs. This is probably related to the non-extensive use of these drugs in this population. Further studies on the use of ICIs as immunological enhancers are needed. In fact, their use has been hypothesized in the treatment of opportunistic or latent infections, reversing T cell immune exhaustion [78].

## 6. Conclusions

ICIs are currently reshaping the landscape of oncologic therapy, opening opportunities for patients previously considered untreatable. PLWHIV have only partially benefited from this revolution, having been excluded from trials investigating the efficacy of these drugs. Nonetheless, due to the growing volume of reassuring data on ICI safety in PLWHIV, currently in those patients with well-controlled infections, it is suggested to employ ICIs as in patients without HIV. Several RCTs are ongoing, enrolling PLWHIV specifically, and more solid data are expected in the near future. The presence of immune exhaustion is a well-described feature of HIV infection, with several pieces of evidence highlighting the role of the immune checkpoints CTLA-4, PD-1, TIM-3, LAG-3, and TIGIT. Nonetheless, the use of ICIs as a therapeutic instrument to eradicate the viral reservoir, despite having been proposed by several authors, is still limited to the in vitro experimental setting and far from use in clinical practice. Finally, due to the lack of ICI use among PLWHIV, it is not possible to draw definitive conclusions regarding the risk of opportunistic infections in this specific population.

## Figures and Tables

**Figure 1 cells-10-02227-f001:**
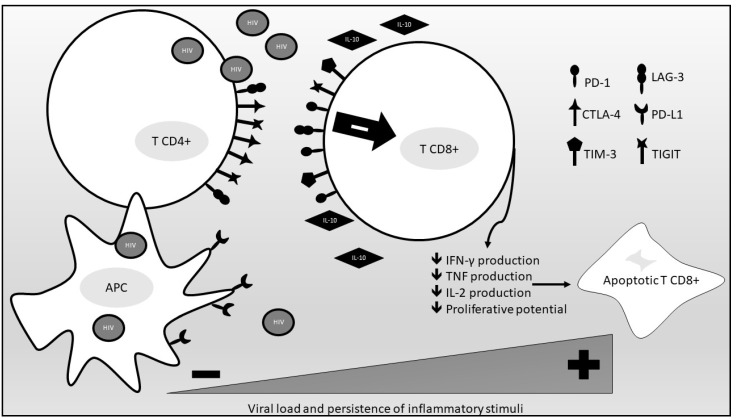
An overview of the immune exhaustion process occurring in PLWHIV. (PD-1, programmed death cell protein-1; CTLA-4, anti-cytotoxic T-lymphocyte-associated protein 4; TIM-3, T cell immunoglobulin and mucin domain-containing protein 3; TIGIT, T cell immunoreceptor with Ig and ITIM domains; LAG-3, lymphocyte-activation gene 3; PD-L1, PD-1 ligand 1; APC, antigen-presenting cell; IL-10, interleukin 10; IFN-γ, interferon gamma; TNF, tumor necrosis factor; IL-2 interleukin 2).

**Table 1 cells-10-02227-t001:** Currently approved ICIs with their indications. (CTLA-4, cytotoxic T-lymphocyte-associated protein 4; PD-1, programmed cell death protein 1; PD-L1, programmed death-ligand 1; FDA, Food and Drug Administration).

Drug	Target	Year of Approval (FDA)	Cancer/s
Atezolizumab	PD-L1	2016	Bladder cancer, breast cancer, liver cancer, lung cancer, and melanoma
Avelumab	PD-L1	2017	Bladder cancer, kidney cancer, and Merkel cell carcinoma
Cemiplimab	PD-1	2018	Squamous cell skin cancer
Dostarlimab	PD-1	2021	Endometrial cancer
Durvalumab	PD-1	2018	Lung cancer
Ipilimumab	CTLA-4	2011	Melanoma, mesothelioma, liver cancer, and lung cancer
Nivolumab	PD-1	2014	Bladder cancer, colorectal cancer, esophageal cancer, gastric cancer, head and neck cancer, kidney cancer, liver cancer, lung cancer, lymphoma, melanoma, and mesothelioma
Pembrolizumab	PD-1	2014	Bladder cancer, breast cancer, cervical cancer, colorectal cancer, cutaneous squamous cell carcinoma, esophageal cancer, head and neck cancer, kidney cancer, liver cancer, lung cancer, lymphoma, melanoma, Merkel cell carcinoma, and stomach cancer

**Table 2 cells-10-02227-t002:** Studies registered on Clinicaltrial.gov assessing ICIs for treatment of cancers in PLWHIV. Search was performed on 24 April 2021. (NSCLC: Non-small-cell lung carcinoma; cART: combination antiretroviral therapy; AEs: adverse events.).

RCT	Study Title	Phase	Cancer	ICI	Primary Objective/s
NCT03354936	ANRS CO24 OncoVIHAC (Onco VIH Anti Checkpoint)	Observational	Any	Any	To evaluate clinical and biological safety of the use of immune checkpoint inhibitors in HIV infected patients with cancer treated by ICIs
NCT04514484	Testing the Combination of the Anti-cancer Drugs XL184 (Cabozantinib) and Nivolumab in Patients WITH Advanced Cancer and HIV	1	Advanced solid tumors	Nivolumab	To determine the safety of combined nivolumab and cabozantinib s-malate (XL184) in PLWHIV with advanced solid tumors To determine the feasibility to deliver the combined nivolumab and XL184 for a minimum of 4 cycles in at least 75% of the subjects or to achieve a confirmed objective response
NCT03304093	Immunotherapy by Nivolumab for HIV+ Patients (CHIVA2)	2	NSCLC Metastatic NSCLC Stage IIIB	Nivolumab	Disease control rate at 8 weeks
NCT02408861	Nivolumab and Ipilimumab in Treating Patients With HIV Associated Relapsed or Refractory Classical Hodgkin Lymphoma or Solid Tumors That Are Metastatic or Cannot Be Removed by Surgery	1	Advanced Malignant Solid Neoplasm Anal Carcinoma Kaposi Sarcoma Lung Carcinoma Metastatic Malignant Solid Neoplasm Recurrent Classic Hodgkin Lymphoma Refractory Classic Hodgkin Lymphoma Unresectable Solid Neoplasm	Nivolumab Ipilimumab	Maximum tolerated dose of nivolumab
NCT03316274	Intra-lesional Nivolumab Therapy for Limited Cutaneous Kaposi Sarcoma	1	Kaposi Sarcoma	Intra-lesional injection of nivolumab	Number of dose limiting toxicity (6 months) Maximum tolerated dose (6 months)
NCT02595866	Pembrolizumab in Treating Patients With HIV and Relapsed, Refractory, or Disseminated Malignant Neoplasms	1	Relapsed, Refractory, or Disseminated Malignant Neoplasms	Pembrolizumab	Frequency of observed adverse events Incidence of immune-related events of clinical interest Incidence of cART-related ECIs of grade 2 or higher AEs
NCT03094286	Durvalumab in Solid Tumors	2	Solid tumors	Durvalumab	Number of HIV patients that receive durvalumab at least during 4 months
NCT04499053	Durvalumab in Combination With Chemotherapy in Virus-infected Patients With Non-small Cell Lung Cancer	2	NSCLC Stage IV	Durvalumab	Adverse events Radiological response

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
