# Peer review of "Immune Checkpoint Inhibitors in People Living with HIV/AIDS: Facts and Controversies"

_cells, 2021, doi:10.3390/cells10092227_

Round 1
Reviewer 1 Report
This review manuscript by Castelli et. al., describes the available literature on cancer immunotherapy using checkpoint inhibitors. In addition, authors describe clinical data on PLWH that have undergone treatment with ICIs. It is an interesting paper worth it for the broader audience but requires extensive clarification and review by a native speaker. In general, I missed a clear distinction between studies performed in untreated HIV infection vs. ART-treated/suppressed individuals and how exhaustion markers (all the ones revised in the paper) recover/are maintained upon ART initiation and in chronic infection. Did any of the studies look at changes in the latent reservoir? My comments are below:
English review throughout the paper, I am only highlighting some examples:
Line 35: In table 1…
Lines 49-52: Not only the English but also it is a long-complicated sentence that requires clarification.
Line 62: objective, aim….
Lines 72-80: mix of English grammar review required
Line 121: sentence order: review
Line 123: review: difficult to understand. Explain and English review
Line 153: exposure
Line 153-157 / 175-177: review English
Line 180: English and scientific name (what items?)
Line 183: English
General comments:
Line 81: A few sentences introducing how ART affects the phenotype and function of “exhausted” cells or cells expressing that phenotype (and specifically what phenotype has been analyzed in different studies is missing.
Lines 76-80: The mechanism need to be explained/clarified
Line 86: clarify what “compromise T-cell homeostasis” refers to
Line 92: what same donors?
Line 101: which are those categories? According to what criteria?
Line 104: what about PD-1/CTLA-4 combined therapy?
Line 114: receiving ART and suppressed? For how long?
Line 127: What are curative HIV targets?
Figure 1: How is the decreased production of different cytokines and proliferative responses affecting apoptosis? Is that the apoptosis of what type of cells? Why there is a disconnection between infected cells and apoptosis? Is that in the context of active replication? If the paper is referring to HIV cure, a figure showing how these phenotypes could affect a cure, then ART-suppression should be considered in the figure.
Line 184: Authors should include a statement on how ICIs could be used in combination with current LRAs towards a cure.
Line 195-198: Clarify, probably English review
Line 200: Where is this conclusion coming from?
Did 2 individuals have detectable viral load after the intervention with ICIs?
Did they just measure plasma viral load? Where there any measures of size of the reservoir?
Line 203-219: All this paragraph should be reorganized to explain the participant’s characteristics first and then the objectives and outcomes. Too many details. Authors should summarize as a whole and interpret the results from the different studies.
Line 209: Not suppressed, just below 200 copies?
Line 211: what was that pre-treatment? Difficult sentence. Review English
Line 219 (reference 51): Why are authors only discussing this paper? What other work is available in the literature? These should be cited
Line 227: Instead of individual numbers and data, authors should summarize and discuss the available data.
Line 229: All this paragraph should be summarized in a few sentences referring to the original papers, and adding discussion from the authors.
Line 242: can authors clarify what are they referring to with upregulated immune system?
Line 247-248: references are missing.
Line 249-289: Between the description of all the opportunistic infections and the lack of cases in PLWH, authors should discuss how these common infections could affect ART-suppressed individuals.
Line 290-293: In what ways? Can authors discuss this further?
Line 296: Limitations of the use of these immunotherapies as a whole paragraph is missing.
Line 303: A more concise description of the clinical trials regarding ART-suppression is required in the manuscript
Reviewer 2 Report
Authors have discussed the potential of immune check point inhibitors (ICI) for HIV elimination in people living with HIV (PLWH). This is a timely review on possible use of ICI as adjunctive therapy in PLWH for HIV eradication. Achieving HIV cure has been a priority and the approach requires a combinatorial treatment strategies, where ICI may have essential role. Authors have reviewed the immunologic status in HIV infection, possible use of ICI as adjuvant therapy for HIV to achieve the elimination of viral reservoirs, safety and effectiveness of ICI, and how the administration of ICIs could impact opportunistic infections in PLWH. All areas to understand the potential of ICI for HIV elimination have been included in a comprehensive manner with relevant reference to the available studies.
There are minor concerns:
The Figure 1 is not clear. It could be elaborate in illustrating the cellular interactions and steps in immune exhaustion, and the regulation of inhibitory signals. The apoptotic cell is not labelled with what type of cell it is, and it is not clear what is the effect on the apoyototic cell. Now it looks like the expression of immune exhaustion markers on T cells is leading to apoptosis of cells. More over, the expression of the exhaustion markers is not only on the HIV infected CD4+ T cells. The figure needs significant improvement.
Round 2
Reviewer 1 Report
This review manuscript by Castelli et. al., describes the available literature on cancer immunotherapy using checkpoint inhibitors. In addition, authors describe clinical data on PLWH that have undergone treatment with ICIs. The reviewed version of the manuscript has substantially improved, and it should be ready for publication albeit some English edits should be reviewed during the editing part of the publication.